# A Work Time Control Tradeoff in Flexible Work: Competitive Pathways to Need for Recovery

**DOI:** 10.3390/ijerph20010691

**Published:** 2022-12-30

**Authors:** Johanna Edvinsson, Svend Erik Mathiassen, Sofie Bjärntoft, Helena Jahncke, Terry Hartig, David M. Hallman

**Affiliations:** 1Department of Occupational Health Sciences and Psychology, University of Gävle, 80176 Gävle, Sweden; 2Institute for Housing and Urban Research, Uppsala University, 75105 Uppsala, Sweden

**Keywords:** occupational health, job autonomy, digitalization, working conditions, working times

## Abstract

Work time control may offer opportunities, but also implies risks for employee recovery, influenced by increased work-related ICT use and overtime work. However, this risk–opportunity tradeoff remains understudied. This study aimed to test two different models of associations between work time control, work-related ICT use, overtime work, and the need for recovery. These models were constructed based on data on office workers with flexible work arrangements. Cross-sectional data were obtained with questionnaires (*n* = 2582) from employees in a Swedish multi-site organization. Regression models treated the three determinants of the need for recovery either as independent, or as linked in a causal sequence. The test of independent determinants confirmed that more work time control was associated with less need for recovery, whereas more ICT use and overtime work were associated with a higher need for recovery. In a test of serial mediation, more work time control contributed to a greater need for recovery through more ICT use and then more overtime work. Work time control also had a competitive, indirect effect through a negative association with overtime work. Our results suggest that work time control is beneficial for employee recovery, but may for some be associated with more work-related ICT use after regular working hours, thus increasing recovery needs. Policies that support work time control can promote recovery, but employers must attend to the risk of excessive use of ICT outside of regular working hours.

## 1. Introduction

Opportunities to recover are central to the maintenance of a health-promoting work environment [1,2]. Yet, recovery has received relatively little attention in occupational health studies, in particular in office settings characterized by flexible work arrangements (FWA). In Sweden, FWA are increasingly common in office-based organizations [3,4,5]. FWA allow employees autonomy to organize when, where, and how they engage in work-related tasks [6,7,8]. A growing body of research on FWA shows conflicting results concerning its effects on health. On the one hand, FWA have been associated with better physical health [9], reduced absenteeism, commitment to work [10], and increased work performance [11]. On the other hand, FWA can reduce well-being [12] in that the work demands of office workers with FWA are characterized by an increased work intensification due to the opportunity to use the flexibility for working more and to use extra effort on their job [5,10,13]. Statistics from the Swedish Work Environment Authority revealed that more than one-third of the Swedish workforce reported to bringing work home and to working overtime to handle their workload [14]. Consequently, FWA may introduce both opportunities and challenges for employee recovery, and in turn to their health [15].

The job demands–resources (JD–R) model categorizes working conditions broadly into demands and resources [16], which can influence health, well-being and performance [17]. The JD–R model suggests that high levels of job demand (e.g., a high workload), along with low levels of resources (e.g., little job control), increase the risk of stress-related ill-health. Conversely, high levels of perceived resources could offset the effects of high demands [16,17]. Accordingly, work time control (WTC) could be considered a resource in FWA, and WTC is thus suggested to promote workers’ health and well-being [18]. WTC refers to ‘an employee’s possibilities to control the duration and distribution of his or her work time’ [19,20]. This implies having control over daily work hours (i.e., starting and ending times and thus the total duration) and time off work (e.g., taking breaks, running private errands during work, and scheduling vacation) [19]. From this point of view, WTC enables employees to organize their time so that work-related efforts are duly balanced with recovery from work [21,22].

However, studies have indicated that WTC could also have negative effects by increasing the work demands, particularly for employees with FWA [5,23,24]. First, WTC could increase the use of information and communication technology (ICT use) for performing work without any temporal or spatial boundaries [8,24,25]. Employees reporting high levels of control over their work have tended to also report high levels of work-related ICT use outside of regular working hours [26]. Thus, employees with high control may remain available for work outside of regular working hours. Reasons for this include, for example, the employee’s own interest, expectations from managers or colleagues, or demands by customers [5,24]. In this context, work-related ICT use has been suggested as obstructing the recovery processes due to difficulties in psychologically disengaging from work tasks [26,27]. Second, employees can also work longer hours. Employees with FWA have reported more overtime work than workers with more scheduled work arrangements [23,28]. Overtime, in turn, has been suggested as a mediator in the association between work-related ICT use outside of regular work hours and recovery [26]. Taken together, a high WTC is in general beneficial for health, but control could also lead to more exposure to work demands, which may challenge recovery by allowing work at times when employees would preferably be recovering from work [23,29].

The effort–recovery theory suggests that periods of work with high demands and stress should be followed by a sufficient recovery to prevent effects that otherwise lead to a considerable need for recovery (NFR) [30]. NFR has been referred to as “a collection of symptoms, temporary feelings of overload, irritability, social withdrawal, lack of energy for new effort, and reduced performance” [31]. NFR is known as an important precursor to psychological overload and health problems [1,32,33]. High NFR has been associated with stress [34], headache, muscle pain [35], increased fatigue [33] and sleep problems [35]. Oppositely, low NFR has shown associations with good general health, mental health, sleep quality and vitality in office workers [34].

A recent study found that work which is flexible in time and location was associated with low well-being and suggested that both stress due to ICT use and NFR may be important mediators of this association [12]. Work-related ICT use may increase NFR, for example, by hindering the psychological detachment from work and thus prolonging the exposure to working demands [25,27,36]. Prolonged exposure to work inevitably leads to reduced time remaining for recovery [15,33]. For example, cross-sectional studies found that expectations of availability or being contacted for work-related matters were associated with increased self-reported physical and mental workload in various occupations [37] and troubles detaching from work at bedtime [38]. Furthermore, previous studies have concluded that overtime work has negative effects on employees’ satisfaction with leisure time [23] and recovery, as seen in indicators of insufficient recovery [15,36,39]. In contrast, an increase in resources such as WTC could, theoretically, help an employee to offset the negative effects of high work demands [17,40]. For example, one such effect would be that overtime work is reduced, and delays in getting recovery during the time it is needed is thereby avoided. This, in turn, may decrease the psychosocial workload caused by and alleviate the negative health effects of extended working hours [41]. 

Little attention has, however, been given to potential determinants of NFR in FWA, and two reviews have called for more knowledge regarding underlying factors linking work characteristics, including control and recovery [26,42]. Furthermore, few large-scale studies have been conducted on homogenous samples of office workers with FWA, especially in the Nordic countries, where FWA are widespread [43] and the prevalence of ICT use is particularly high [8]. Thus, it is still unclear how and to what extent WTC, work-related ICT use, and overtime work are associated with NFR in office workers with FWA. Conceivably, WTC, work-related ICT use outside of regular working hours, and overtime work may have important independent associations with NFR. Yet, the three determinants may also act together, as shown above, with increased opportunities for extensive work due to work-related ICT use, allowed by the extended WTC, which in turn allows for such work flexibility. Therefore, this cross-sectional study aimed to test two different models of associations between WTC, work-related ICT use, overtime work, and NFR, each based on data on office workers with FWA. We thus add to the existing theory and research by providing a better understanding of the tradeoff between effects of WTC that may serve or undermine a balance between work and recovery in FWA, in particular, by proposing potentially competitive pathways [44].

The first model emphasizes the independent associations of each of the three determinants with NFR. Drawing on the JD–R model and the effort–recovery theory, we expect that WTC will act as a resource and therefore be associated with less NFR. We also expect that a higher extent of work-related ICT use and more overtime work will act as job demands and therefore be associated with a greater NFR. The first model includes the following hypotheses:

**Hypothesis** **1a.***More WTC will be associated with less NFR*.

**Hypothesis** **1b.***More work-related ICT use will be associated with more NFR*.

**Hypothesis** **1c.***More overtime work will be associated with more NFR*.

Model 2 includes the effects specified in Model 1 but situates them in a pattern of multiple indirect effects, including serial mediation by work-related ICT use and overtime work in the association between WTC and NFR. We expect that WTC will have a direct effect on NFR, as well as three indirect effects. First, we expect that high levels of WTC will be associated with a greater NFR through a higher extent of work-related ICT use, but less NFR through fewer hours of overtime work. Further, we expect that higher levels of WTC will be associated with a higher extent of work-related ICT use, leading to more overtime work, and in turn, a greater NFR. However, we acknowledge the plausibility of other directions of the associations and mediations. Model 2 tests the following hypotheses:

**Hypothesis** **2a.***Work-related ICT use will mediate the association between WTC and NFR (positive indirect effect)*.

**Hypothesis** **2b.***Overtime work will mediate the association between WTC and NFR (negative indirect effect)*.

**Hypothesis** **2c.**
*Work-related ICT use and overtime work, in serial, will mediate the association between WTC and NFR (positive indirect effect).*


## 2. Methods

### 2.1. Design

The study had a cross-sectional design and used data from the research project, “Flexible work: health-promoting interventions for sustainable digitalized work”, conducted at the Swedish Transport Administration (STA), a governmental agency in Sweden. The research project has been described previously [40,45].

### 2.2. Procedure and Sample

Data collection took place in October 2016, when a comprehensive web-based questionnaire was sent out by e-mail to all employees and managers with flexible work arrangements (i.e., flextime or non-regulated working hours; *N* = 4900). To increase participation, the employer allowed employees to answer the questionnaire during regular working hours. They received four reminders at two-week intervals. In total, 3259 responded (response rate 66.5%). After exclusion of 284 part-time workers, consultants, and employees with ongoing sick leave or ongoing parental leave, the sample of full-time office workers was *n* = 2975. Of these, 2678 had complete data on the determinant variables whereas 297 had not. Some did not provide data for all variables in the analyses, leaving *n* = 2582 for the tests of the full models with adjustments for covariates, as detailed below.

### 2.3. Measurements

Questions not originally in Swedish were translated into Swedish and then back translated into English by the authors, one of whom is native English speaker. To adapt the questionnaire to the organization, we had discussions with a reference group consisting of both researchers and experts on surveys within the STA. Furthermore, the items were validated in a pilot study conducted prior to the data collection. First, by qualitative think-aloud interviews with office workers from the STA with experience in survey development and/or FWA (*n* = 3). Second, by quantitative tests of the questionnaire on employees from three other organizations than the STA, who had FWA or scheduled work arrangements (*n* = 266). Overall, the results indicated sufficient content, face, and construct validity [46]. The questions solicited participant experiences over the last three months.

Work time control. We measured WTC using two items, modified from Allvin and colleagues [3]. These were, “I can control what times I work on a certain day” and “I can control when to work with different tasks” (author translations here from the original Swedish). The scale ranged from 0 = completely incorrect to 4 = completely correct. The two items were averaged to create an index where higher values indicate a higher degree of WTC (Spearman–Brown coefficient = 0.76).

Work-related ICT use. We used five items, customized within the research group, to measure work-related ICT use outside of regular working hours: “Do you work outside of regular working hours with the following: Reading and/or writing e-mails?; Reading and/or writing text messages?; Performing work tasks on the computer?; Calling and/or answering phone calls via smartphone?; Calling and/or answering calls via Skype?”. Five answer alternatives were available for each item: from 0 = not at all to 4 = to a very high degree. Answers to the five items were averaged to create an index where higher values correspond to a greater extent of work-related ICT use (Cronbach’s alpha = 0.80).

Overtime work. Overtime work was measured using two items asking for the participants’ contracted work hours and their actual work hours per week during the last three months. Similar to Bjärntoft and colleagues [40], we calculated the difference between the contracted and the actual work hours, so that higher positive values indicate more overtime work.

Need for recovery. We used the need for recovery scale [32], which includes 11 items in a dichotomous response format (no = 0, yes = 1). Example items are, “By the end of the working day, I feel really worn out” and, “Often, after a day’s work I feel so tired that I cannot get involved in other activities”. The 11 individual scores were added up and transformed into a scale ranging between 0 and 100, where higher scores indicate a higher NFR (Cronbach’s alpha = 0.86). Employees giving at least 6 positive answers, i.e., scoring ≥ 6/11 or 54.5%, were considered to have a high NFR, and respondents scoring less than this were considered to have a low NFR. This cut-off point has been proposed in previous studies, suggesting that employees with a ‘high’ NFR have a greater risk of developing physical and/or psychological health complaints [31,34,47].

Covariates. We included several covariates in our analyses to reduce the risk of biased associations between the variables of interest. Covariates were selected based on the literature, with a likely relation to either WTC or NFR, i.e., gender (men or do not want to categorize = 0, women = 1), age (years), and children at home (no children at home = 0, one or more children at home = 1) [48].

Demographic variables. We included questions about work arrangements (non-regulated workhours, flextime and other) and a question about seniority (years employed in the organization).

### 2.4. Data Analysis

All statistical analyses were performed using SPSS software version 24 (IBM, USA). Confirmatory factor analysis (CFA) was used to verify the factor structure of each variable before creating summary indices of variables. The study hypotheses were tested using multiple linear regression analysis. We assessed potential clustering of data by work location and occupation using intraclass correlations (ICC). Only 0.4% and 1.5% of the variance in NFR were found to be related to clustering, respectively, while the remaining variance was attributed to intersubject variation. Therefore, we decided not to include occupation and location in the regression models. After checking descriptive statistics and variable skewness, we examined the Pearson correlation between the study variables. The residuals were visually inspected for normality and no marked deviations were observed. No multivariate outliers were identified using the criterion: leverage > 1. The sample was sufficiently large to minimize regression model bias in estimates and confidence intervals [49].

We tested two different models with WTC, work-related ICT use, and overtime work as potential determinants of NFR, as described in the aims. For Model 1, associations were examined using multiple linear regression analysis, with the independent effect of each determinant appearing after adjustment for the effects of the other two determinants. For Model 2, the direct and indirect effects of WTC were analyzed in a serial multiple mediation model using Hayes’ Process macro for SPSS [50], as shown in Model 6. We tested work-related ICT use and overtime work in the association between WTC and NFR, both as single mediators (H2a, H2b) and in the specific order outlined above (H2c). This order assumes that more WTC allows for additional work-related ICT use outside of regular working hours and, in turn, more overtime work, with negative consequences for NFR. Confidence intervals (95% CI) for the effects in the mediation models were estimated by a bootstrapping technique. All regression models were run first without and then with adjustment for the covariates age, gender, and children at home.

## 3. Results

### 3.1. Descriptive Statistics

Sample demographics are shown in Table 1. Men were more frequently found among participants than women. Ages ranged from 21 to 73 years. Furthermore, around half of the participants had one or more children under 18 years old in their household. About one third, or 32.9%, of the participants, reported a ‘high’ NFR.

Bivariate correlations between the variables in the analyses are given in Table 2. Rated as ‘small’ (0.10 to 0.29), ‘medium’ (0.30 to 0.49) and ‘strong’ (0.50 to 1.0) [51] (p. 79–81), all correlations were, at the most, small, except for the medium-size, positive correlation between work-related ICT use and overtime work (*p* < 0.001). Overall, the correlations indicate no potential problem concerning collinearity.

### 3.2. Independent Associations of Work Time Control, Overtime Work, and Work-Related ICT Use with Need for Recovery

In Model 1, we expected WTC to be associated with NFR (H1a), work-related ICT use to be associated with NFR (H1b), and overtime work to be associated with NFR (H1c). Overall, the results align with all three hypotheses. WTC was negatively associated with NFR, while work-related ICT use and overtime work were positively associated with NFR (all *p* < 0.001). The unadjusted model (Model 1a in Table 3) indicated that a 1-unit increment in WTC, work-related ICT use, and hours of overtime work were associated with statistically significant changes in NFR; these were, respectively, around a 7 percent reduction in and 4 and 1 percent increase in NFR. Estimates and 95% CIs were similar when controlling for age, gender, and children at home (Model 1b in Table 3), and the variance explained showed a marginal increase from 9% to 11% after adding the covariates. Gender (i.e., being a woman) had a significant positive association with NFR (*B* = 6.85), whereas age and children at home had not. The results of Model 1b as related to Hypotheses 1a–c are given in Figure 1.

### 3.3. Serial Multiple Mediation Model of the Association between Work Time Control and Need for Recovery

In Model 2, we expected that WTC would have a direct effect on NFR as well as three indirect effects: through work-related ICT use as a single mediator (H2a), through overtime work as a single mediator, (H2b), and through both work-related ICT use and overtime work in serial mediation (H2c). Table 4 reports the coefficients for the total, direct and indirect effects, adjusted for age, gender, and children at home.

We found significant indirect effects of the two candidate mediators when considered separately and in serial. In itself, work-related ICT use appeared to transmit a positive indirect effect. In keeping with H2a, more WTC was associated with more work-related ICT use outside of regular working hours, such that a one-unit change in WTC produces a 0.03 unit change in work-related ICT use outside of regular working hours, which in turn was associated with greater NFR. In line with H2b, overtime work itself had a negative indirect effect, in that more WTC was associated with a reduction in overtime work after an adjustment for the other indirect effects. In keeping with H2c, the analysis revealed a significant negative indirect effect of WTC on NFR through the two mediators in serial. The indirect effect of work-related ICT use outside of regular working hours, through overtime work, showed a one-unit change associated with a 2.31 h increase in overtime work per week, and each unit of overtime work was then associated with a 0.88 unit increase in NFR.

The total indirect effect was, however, not significant. This can, in part, be explained by competitive mediation, in which the one negative indirect effect (WTC leading to less overtime work and in turn lower NFR) offset the two positive indirect effects [44]. Furthermore, the direct effect of WTC on NFR remained significant and negative, and in the same order of magnitude as in Model 1, after adjustment for the indirect effects. This suggests that WTC serves to reduce NFR mainly through processes other than those represented by the mediators assessed here. Adding age, gender, and having children at home as covariates to the model did not notably change the results found in the unadjusted model (Model 2b in Table 4). The results of Model 2b as related to Hypotheses 2a–c are given in Figure 2.

## 4. Discussion

This cross-sectional study aimed to test two different models of associations between WTC, work-related ICT use, overtime work, and NFR based on data on office workers with FWA. The first model used a conventional multiple linear regression approach which established that the three determinants had significant independent associations with NFR. The second model affirmed these associations but also indicated the plausibility of indirect effects, although some of these appeared weak. Taken together, the results indicate that WTC is involved in two conflicting scenarios. On the one hand, WTC may support recovery processes by allowing a balance between work-related efforts and recovery resulting from individual needs. On the other hand, WTC may lead to more work-related ICT use outside of regular working hours, more overtime work, and higher NFR. The opposite signs of these two pathways suggest that WTC is beneficial for employee recovery, but that it may also lead to more work-related ICT use outside of regular working hours, which could then increase recovery needs.

Overall, our analysis confirmed the expected independent associations for all three determinants, with a joint effect size of 9% explained variance. The negative association between WTC and NFR follows our expectations (H1a), considering WTC to be facilitating recovery. This view draws support from the JD–R model and empirical findings that WTC enables employees to switch between work-related tasks and recovery following individual needs [17,22]. In line with H1b, we found a positive association between work-related ICT use and NFR. Our findings indicate that employees working with ICT outside of regular working hours report a higher NFR, consistent with findings in previous studies [12,52]. However, we emphasize that not only the ICT use per se, but also other factors that may be associated with work-related ICT use outside of regular working hours, can be important for the NFR. This includes factors such as job demands [33], availability expectations [24], and/or a lack of psychological detachment [53]. Moreover, we found a positive association between more overtime work and more NFR (H1c). This agrees with previous research, although studies of overtime work and recovery have mainly focused on samples of shift workers in different occupations [41,54]. The covariates in the model did not change the estimates in focus with our hypotheses, and the adjusted model only explained an additional 2% of the variance in NFR. Thus, substantial confounding by age, gender, and children at home can be ruled out, even though being a woman was associated with an about 7% higher NFR. Whether this effect is comparable to that in other studies of office workers with FWA is not known due to the lack of studies.

As single mediators, work-related ICT use had a positive indirect effect (H2a), and overtime work had a negative indirect effect on the association between WTC and NFR (H2b), as expected. These findings follow the idea that WTC on the one hand may lead to more work-related ICT use [24,26], while it, on the other hand, may prevent overtime work. Specifically, the mediating pathway between WTC and NFR through overtime work (H2b) represented a competitive mediation in the association between WTC and NFR. A previous study reported positive associations between overtime work and higher NFR, but only for employees who experienced high job demands [55]. This may help in explaining our findings, in that WTC enables some of the employees to adapt their time to fit recovery needs [21].

Furthermore, the serial pathway from WTC through ICT use and overtime work appeared to have only a weak indirect effect on NFR (H2c). The small effect is mainly explained by the first step of the putative causal sequence, in which WTC showed a weak association with increased work-related ICT use outside of regular working hours. However, the subsequent links to overtime work and NFR showed larger values. This result implies that there may be a subgroup of ICT users which does not benefit from a work situation with high WTC [24,26]. The organization of work hours may vary due to, for example, different strategies for combining work and leisure [56], individual preferences of boundary management or boundary control (e.g., integrating or separating work and private life) [57], and differences in the reasons for working flexibly [58]. Conceivably, the relationship between WTC and ICT use may be stronger and positive in some groups, stronger and negative in other groups, and weak in still other groups.

Overall, there is a general lack of studies on NFR in employees with FWA, a reality which precludes a direct comparison of our results with similar studies and a clear prediction of the health effects of the observed levels of NFR, including differences between subgroups. However, the average NFR in the present study appeared higher than that in previous studies of office workers [34] and other occupations [1]. Approximately, one third of the employees reported to experience a ‘high’ NFR, a development which could have implications for health [31]. However, further investigations into this was beyond the scope of this study, but does represent an issue for future research studies.

### 4.1. Theoretical and Practical Implications

Our findings make a theoretical contribution to research by showing two opposite directions of the association between WTC and NFR, and that work-related ICT use and overtime work may be influential as mediators in this relationship, both individually and in serial. These results contribute to a better understanding of the tradeoff between positive and negative effects of WTC, and point to the need for further research on how to promote recovery and health in flexible work. Furthermore, the results can be used as a basis for further theoretical development and testing in prospective studies and interventions to promote recovery in FWA, including a more detailed understanding of the needs of different ‘types’ of employees. The link between increased NFR and long-term negative health effects is well established; however, the results of the associations in this study provide new insights into a plausible causal order among the determinants of NFR.

The findings have valuable practical implications for the development of, for example, workplace interventions, policies, and guidelines with a focus on promoting recovery and health in organizations offering FWA. The results suggest that organizations should enable WTC, while simultaneously protecting against excessive work-related ICT use and overtime work in order to ensure recovery from work. For example, employers could develop policies to clarify expectations on availability outside regular working hours and ensure that the workload is reasonable to actually enable the employees to set or maintain their preferred boundaries regarding work-related ICT use and overtime work. This would likely lead to a promotion of recovery and maybe even health in employees with FWA.

### 4.2. Strengths and Limitations

The main strengths of this study are the high response rate (66.5%) and the large sample, drawn from a homogeneous population of office workers with FWA. Furthermore, all measures were validated in a pilot study, resulting in good content, construct and face validity [46]. Moreover, our study considered several technological devices and ways to work with ICT (for example working with e-mails, text messages, phone calls, or on the computer), in contrast to previous studies that mainly focused on smartphone use [59,60,61].

However, the study also has several limitations. The cross-sectional design implies that associations should not be interpreted as causal, and that reversed causality may be present. We acknowledge that cross-sectional data is not optimal for investigating processes [62,63]. However, our models can examine the validity of theoretical claims. The results can then be used as basis for a more optimal research model to further investigate associations, i.e., a model based on longitudinal data. Also, all measures are based on self-reports, which are known to be subject to various biases [64]. For example, there is a risk of common method variance that could lead to overestimated associations. Further, the sample was recruited from one organization, which may limit generalizability to the general working population. As the scores of work-related ICT use outside of regular working hours were relatively low, the results would likely be different in an organization with a higher extent of work-related ICT use. On the other hand, the sample represents a variation in work tasks, education, gender, age, and organizational position, which could support the generalizability of the results for employees with FWA.

Regarding the measures, WTC as measured here only taps the dimension control over daily hours (i.e., when to start and end the workday); however, the concept of WTC also includes a dimension of control over when to take time off work (i.e., the control over when to take breaks or running private errands during a workday, scheduling vacation and/or other types of leave) [60]. Our focus here follows from the recommendation by Albrecht and colleagues [19] to distinguish between these dimensions. Furthermore, the measurement of NFR has focused on work-related fatigue at the end of, or just after, a workday [31]. This may be problematic in the context of FWA, in which the start and end times of work may be difficult to define [65] due to the grey boundaries between work and private life [66]. Accordingly, even though NFR has been considered a valid measurement to determine the NFR in workers from different occupations [2], we acknowledge that other options may be more suitable for measuring recovery in office workers with FWA.

### 4.3. Suggestions for Further Research

Research on recovery should further investigate factors permitting and promoting employee recovery in FWA, as one third of the office workers in this study reported experiencing a high NFR. We found WTC to be negatively associated directly with NFR, and at the same time positively associated indirectly via work-related ICT use, which gives rise to issues regarding differences between employees when facing the same work demands and the same resources in FWA. Hence, it would be valuable to identify potential subgroups in which work-related ICT use outside of regular working hours may be particularly common, which could be helpful to more narrowly specify the practical implications of the results.

Another contribution would be to identify other factors in FWA that may contribute to greater NFR, either as potential determinants, mediators or moderators. Although our results confirmed the expected associations in the present study, we suggest that other potential determinants and mediators besides WTC, work-related ICT use, and overtime work may be influential for the NFR in office workers. Examples of individual factors may include psychological detachment, commitment to work, age, and gender; group factors may be such factors as expectations of availability; factors at an organizational level may include leadership and workload.

The findings from this study provide a basis for further research, and we emphasize the need for longitudinal studies of FWA with objective measurements of potential determinants and recovery outcomes. For example, more knowledge on longitudinal associations, including measurements of the actual work-related ICT use, working hours, and objective indicators of recovery (e.g., heart rate variability), could help in drawing causal inferences about the effect of these variables, as well as the long-run health and work productivity of employees.

## 5. Conclusions

This cross-sectional study aimed to test two different models of associations between work time control, work-related ICT use, overtime work, and need for recovery based on data from office workers with flexible work arrangements. We found that employees experiencing a higher degree of control over their flexible work time had a decreased need for recovery. However, the more work-related ICT use and overtime work increased, the more the need for recovery increased. Overall, these results suggest that, when working flexibly, a high degree of work time control is beneficial for employee recovery. However, work time control may also increase recovery needs if work-related ICT is used outside of regular working hours and overtime work are extensive. Policies that support work time control may provide a framework for work that facilitates recovery. However, employers must also regularly evaluate if the workload is too high and attend to the risk of excessive use of ICT outside of regular working hours. This, in combination with support for individual boundary management, might reduce the negative effects on the need for recovery in flexible work arrangements. However, these interventions need to be evaluated further.

## Figures and Tables

**Figure 1 ijerph-20-00691-f001:**
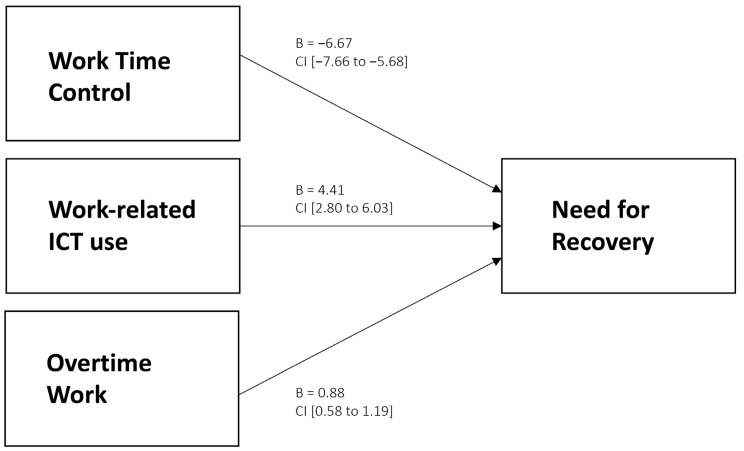
Associations from the multiple linear regression analysis of relations between the determinants work time control, work-related ICT use, and overtime work and the dependent variable need for recovery. The B-values (95% confidence intervals) show the unstandardized coefficients adjusted for age, gender, and children at home.

**Figure 2 ijerph-20-00691-f002:**
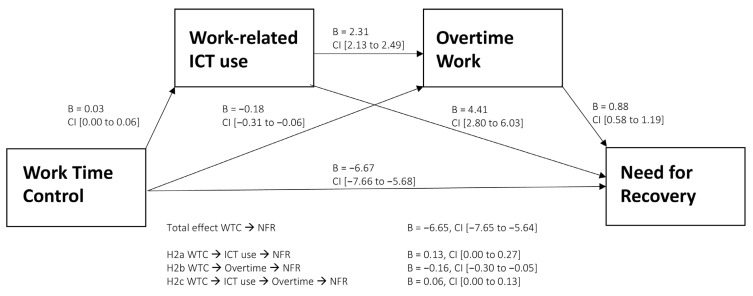
Results of the serial mediation test of the relationship between work time control and need for recovery, through work-related ICT use outside of regular working hours and overtime work in simple and serial indirect effects. The B-values (95% confidence intervals) show the unstandardized coefficients after adjustment for age, gender, and children at home. Values for direct associations between the factors are presented in the figure. The coefficients for the three indirect effects, presented below the figure, align with their respective hypotheses.

**Table 1 ijerph-20-00691-t001:** Sample demographics. The values show the number (*n*) and percentage of respondents (with *n* = 2975 being 100%) for categorical variables and mean values with standard deviation (*SD*) for scales.

Variables	*n*	%	Mean (*SD*)
Age (years)	2915		48.1 (9.5)
Gender	2894	97.3	
Women	1139	38.3	
Men	1671	56.2	
Do not want to categorize	84	2.8	
Children at home	2975	100	
No children	1364	45.8	
One or more children at home	1611	54.2	
Work arrangement	2959	99.4	
Non-regulated workhours	2080	69.9	
Flextime	845	28.4	
Other	34	1.1	
Seniority (years)	2926		14.7 (11.0)
Work time control (scale 0–4)	2963		2.7 (1.2)
Work-related ICT use (scale 0–4)	2967		1.2 (0.8)
Overtime work (hours per week)	2698		2.7 (4.1)
Need for recovery (scale 0–100)	2963		35.9 (30.5)

**Table 2 ijerph-20-00691-t002:** Pearson correlations between study variables.

	Variables	1	2	3	4	5	6	7	n
1	Need for recovery	-	−0.23 **	0.14 **	0.17 **	−0.08 **	0.12 **	0.03	2963
2	Work time control		-	0.04 *	−0.04 *	−0.08 **	0.02	0.03	2952
3	Work-related ICT use			-	0.43 **	0.03	−0.02	0.12 **	2955
4	Overtime work				-	−0.08 **	−0.04 *	−0.01	2691
5	Age					-	−0.16 **	−0.23 **	2912
6	Gender						-	0.02	2890
7	Children at home							-	2963

* *p* < 0.05, ** *p* < 0.001.

**Table 3 ijerph-20-00691-t003:** Associations from the multiple linear regression analysis of relations between the determinants work time control (WTC), work-related ICT use, and overtime work and the dependent variable need for recovery (NFR), without (Model 1a) and with adjustment (Model 1b) for age, gender and children at home. Unstandardized and standardized coefficients are reported.

	Model 1a							Model 1b						
	*n* = 2678							*n* = 2582						
Variables	*B*	*SE*	β	*t*	CI (95%)	*R*	*R2*	*B*	*SE*	β	*t*	CI (95%)	*R*	*R2*
Intercept	46.24	1.74	---	26.53	42.82 to 49.66	0.30	0.09	57.71	1.74	---	15.59	50.45 to 64.97	0.34	0.11
Work time control	−6.45	0.51	−0.24	−12.67	−7.46 to −5.46			−6.67	0.51	−0.25	−13.20	−7.66 to −5.68		
Work-related ICT use	4.26	0.82	0.11	5.17	2.64 to 5.87			4.41	0.82	0.11	5.37	2.80 to 6.03		
Overtime work	0.82	0.16	0.11	5.21	0.51 to 1.12			0.88	0.16	0.12	5.67	0.58 to 1.19		
Age								−0.29	0.06	−0.09	−4.61	−0.41 to −0.16		
Gender								6.85	1.18	0.11	5.79	4.53 to 9.17		
Children at home								−0.56	1.18	−0.01	−0.47	−2.87 to 1.76		

Note: Gender: Men or do not want to categorize = 0, Women = 1; Age: Years; Children at home: No children = 0, Children = 1.

**Table 4 ijerph-20-00691-t004:** Total, direct and indirect effects estimated for the serial mediation model relating work time control (WTC), work-related ICT use, and overtime work (OW) to need for recovery (NFR), first without (Model 2a) and then with adjustment (Model 2b) for age, gender and children at home. Unstandardized and standardized coefficients are reported.

		Model 2a				Model 2b			
		*n* = 2678				*n* = 2582			
	Mediation Paths	*B*	β	*t*	CI (95%)	*B*	β	*t*	CI (95%)
Intercept		53.74				57.71			
Total effect									
	WTC → NFR	−6.42	−0.24	−12.65	−7.41 to −5.42	−6.65	−0.25	−12.92	−7.65 to −5.64
Direct effect									
	WTC → NFR	−6.42	−0.24	−12.84	−7.40 to −5.44	−6.67	−0.25	−13.20	−7.66 to −5.68
Indirect effects									
	Total	0.0002	0.000		−0.22 to 0.22	0.03	0.001		−0.20 to 0.26
	WTC → ICT → NFR	0.12	0.004		0.01 to 0.25	0.13	0.005		0.00 to 0.27
	WTC → OW → NFR	−0.18	−0.007		−0.32 to −0.06	−0.16	−0.006		−0.30 to −0.05
	WTC → ICT → OW → NFR	0.06	0.002		0.00 to 0.12	0.06	0.002		0.00 to 0.13

Note: Gender: Men or do not want to categorize = 0, Women = 1; Age: Years; Children at home: No children = 0, Children = 1.

## Data Availability

The dataset analyzed during the current study is available from the corresponding author upon reasonable request.

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
