# Peer review of "A Work Time Control Tradeoff in Flexible Work: Competitive Pathways to Need for Recovery"

_ijerph, 2022, doi:10.3390/ijerph20010691_

Round 1

Reviewer 1 Report

The topic addressed is an interesting one. The relationship between work and leisure is much less popular than the work-family relationship. It is good that such a text was created.

The theoretical justification does not raise my doubts.

For that, the method part, in my opinion, needs significant changes.

Hypotheses 1a-1c are formulated very generally. They only talk about correlation, they do not give its sign (positive vs. negative). This conflicts with the content of their justification (rows: 100-104), where directions are given (WTC+ to NFR-). Unfortunately, the content of the justification for the existence of the assumed relationships is very poor. If there is no theoretical basis for the hypothesis then it cannot be made. There must be a premise. The mechanism of inference by analogy can be used, but there must be justification. In the text (rows: 108-116), hypotheses are made about mediation, which, with a single study, is unacceptable without indicating why one variable mediates the relationships of other variables. I find it difficult to understand why, according to the Authors, WTC explains ICT use?

If there is no rationale then you have to conduct exploratory research without formulating hypotheses, but then you can't test models that are hypotheses of relationships either.

The problem is all the more important because the calculated model does not test simple relationships (correlations) but dependencies (Explanatory and Explained Variables) and even more mediation models.

Testing mediation models requires either longitudinal studies or a good theory justifying the order of variables in the model. Here there is neither one nor the other. For what it's worth, line 124 mentions that the research was longituidal. However, this is not reflected in the data and analysis. Hypothesis 2c includes hypotheses 2a and 2b so I don't know why they are put forward. 

There is a loss of about 300 respondents in line 136, but there is no stated reason for this sample reduction. 

It would also be useful to have a cultural adaptation of the tools used and not just a back translation. 

The list of controlled variables is poor. To me, the question about seniority was missing.

In Table 1. the numbers of respondents appear inconsistent with the description in section 2.2.

Figure 1 illustrates Model 1a but includes parmeters from the Table describing Model 1b. The illustration of path strength using the B parameter is not very clear. Beta shows better which variables are more strongly related and which are less strongly related.

At line 287, the authors claim to have conducted a cross-sectional study, which is not reflected in the paper presented. It's hard to talk about cross-sectional because there are no between-group comparisons. Models are described but not compared.

Reviewer 2 Report

This is a very interesting study, especially in the postcovid era of hybrid work development. On the other hand, there are two inaccuracies. Firstly: "Data collection took place in October 2016" - in the introduction, but also in the theoretical part, definitely more recent publications are cited as the basis for the research, so something doesn't add up here. Maybe there is an error in the year of the research, because the 2016 research in the context of more recent studies and report data does not correspond a bit. The second comment concerns the sources, which are very numerous, but there are relatively few of the most up-to-date ones. Also, some of the items are duplicated, e.g. 30 and 35 (the doi is the same but the year differs), or 27 and 49. I suggest checking and correcting the references. The age category of the respondents is not entirely clear to me, as only the minimum and maximum age is given, there is no internal structure to infer the meaning of age, its ranges, generational groups.

Round 2

Reviewer 1 Report

I thank the authors of the text for their response.

I consider most of the comments clarified and the corrections sufficient.

However, I still do not see , contrary to the statement in the response, the rationale for the relationship between WTC and ICT. The cited literature does not point to the increased use of ICT only to the accompanying use of ICT exceeding working hours. Either a better justification or a change in the model is needed.
